# Unilateral Strength Training Imparts a Cross-Education Effect in Unilateral Knee Osteoarthritis Patients

**DOI:** 10.3390/jfmk7040077

**Published:** 2022-09-28

**Authors:** Warren Bowen, Ashlyn K. Frazer, Jamie Tallent, Alan J. Pearce, Dawson J. Kidgell

**Affiliations:** 1School of Exercise and Nutrition Sciences, Deakin University, Melbourne 3125, Australia; 2College of Science, Health and Engineering, La Trobe University, Melbourne 3086, Australia; 3Monash Exercise Neuroplasticity Research Unit, Department of Physiotherapy, Monash University, Melbourne 3800, Australia; 4School of Sport, Rehabilitation and Exercise Sciences, University of Essex, Colchester CO4 3SQ, UK

**Keywords:** cross-education, strength training, knee osteoarthritis, pain, functional performance

## Abstract

**Background:** Worldwide, 86 million individuals over the age of 20 were diagnosed with knee osteoarthritis (KOA) in 2020. Hallmark features of KOA are the loss in knee extensor strength, increasing knee pain severity, and deficits in functional performance. There is a critical need for the investigation into potential cost-effective therapeutic interventions in the treatment of KOA. A potential therapeutic option is the cross-education phenomenon. **Methods:** This was a non-blinded randomized control trial, with a 4-week intervention, with a pre, post and follow-up assessment (3 months post intervention). Outcome measures of isometric knee extensor strength, rectus femoris muscle thickness and neuromuscular activation were assessed at all-time points. **Results:** Compared to age-matched KOA controls, 4 weeks of unilateral strength training in end-stage KOA patients increased strength of the untrained affected KOA limb by 20% (*p* < 0.05) and reduced bilateral hamstring co-activation in the KOA intervention group compared to the KOA control group (*p* < 0.05). **Conclusions:** A 4-week-long knee extensor strength training intervention of the contralateral limb in a cohort with diagnosed unilateral KOA resulted in significant improvements to knee extensor strength and improved neuromuscular function of the KOA limb. Importantly, these results were maintained for 3 months following the intervention.

## 1. Introduction

The loss of knee extensor strength is a hallmark feature of knee osteoarthritis (KOA), which is of critical importance, as knee extensor strength is a key determinant of avoiding functional disability during the progression of KOA [1]. Atrophy of the knee extensors partially explains the loss in knee extensor strength during the progression of KOA [2,3]. However, the primary cause of knee extensor strength loss appears to be arthrogenic muscle inhibition (AMI) which results in the inability of the nervous system to completely activate the knee extensors [4,5]. In unilateral KOA the knee extensor strength deficit is a bilateral occurrence, with the contralateral limb affected to a lesser extent [4]. Knee extensor strength deficits of the KOA limb are significant; the KOA limb demonstrates deficits between 36–48% during end-stage KOA when compared to healthy age-matched controls [4]. Interestingly, studies frequently use the contralateral limb as a control [6,7] demonstrating strength deficits in the KOA limb ranging from 18–31% when compared to the contralateral limb. However, some caution needs to be taken when using the contralateral limb as a control, as it does not represent normal knee extensor strength when compared to healthy age-matched controls, with knee extensor strength deficits ranging from 17–36% [7,8,9].

The cross-education phenomenon is a neurological response to a unilateral strength training stimulus that results in a strength increase to both the trained and untrained contralateral homologous muscle group [10], in the absence of changes to muscle morphology [11]. A recent meta-analysis reported that in young healthy adults the mean strength increase of the untrained lower limb is 16.4% [12]. The proposed neurological mechanisms that underpin the cross-education effect appear to be driven by changes in the excitability of the primary motor cortex ipsilateral to the trained limb [10]. Specifically, increased ipsilateral corticospinal excitability, reduced corticospinal inhibition, reduced short-interval cortical inhibition and potentially reduced interhemispheric inhibition [10] accompany the changes in strength of the untrained limb. The application of cross-education as a clinical exercise therapy has been successfully trialed in limb immobilization [13,14,15,16], forearm fractures [17], bilateral KOA [18], unilateral KOA [19], multiple sclerosis [20] and stroke [21]. The application of cross-education in unilateral KOA appears to have merit; however, previous studies [18,19] did not examine the potential neuromuscular mechanisms that underpin the cross-education effect.

Increasing knee extensor strength of both the affected KOA and unaffected contralateral limbs in unilateral KOA towards the levels seen in healthy age-matched controls is of critical importance in reducing functional disability. The contralateral limb is the dominant predictor of functional performance; therefore, an increase in strength of this limb may be beneficial in improving functional performance. Targeting the contralateral unaffected limb will potentially bypass any acute aggravation from heavy load strength training of the affected KOA limb. Using this approach (i.e., cross-education) is likely to increase knee extensor strength of both the trained (unaffected KOA limb) and untrained affected KOA limb. Decreased knee extensor strength has also been previously highlighted as a risk factor for the development of KOA in the contralateral limb [22]. Again, improving knee extensor strength of this limb may provide long term benefits to attenuating or slowing the progression of KOA.

Decreased knee extensor strength throughout the progression of KOA is primarily due to the inability of the nervous system to completely innervate a muscle, which is termed central activation deficit (CAD) [23]. While the loss of muscle cross-sectional area (CSA) has also been implicated in knee extensor strength loss in KOA, CAD appears to account for more than double the deficits attributed to muscle atrophy in KOA [24]. In addition, hamstring co-activation has also been implicated as being involved in reduced knee extensor strength in KOA. It has been speculated that increasing levels of co-activation aid in joint stability, at the cost of knee extensor strength [25]. Conversely, mixed results have been reported, with increases in co-activation during maximal isometric knee extension [25], and no differences in KOA co-activation when compared to a healthy control group [26]. However, to date, no study has investigated the influence of a cross-education intervention on hamstring co-activation during maximal isometric contractions.

Therefore, the aims of this study were to investigate the effects of 4 weeks of unilateral strength training of the contralateral limb (unaffected limb) in individuals with unilateral KOA, compared to untrained individuals with unilateral KOA and a healthy age-matched control on knee extensor strength and neuromuscular activation. It was hypothesized that unilateral strength training of the unaffected contralateral limb in unilateral KOA would increase knee extensor strength bilaterally, imparting a cross-education effect to the untrained affected KOA limb. Further, the improvements in knee extensor strength would be retained in the three-month period following the intervention. It was also hypothesized that unilateral strength training of the unaffected contralateral limb in participants with unilateral KOA would decrease co-activation of the hamstring muscle group, underpinning any changes in strength in the trained and untrained limbs. Given that the reported neural mechanisms modulating the cross-education effect, the above hypotheses appear to be supported by the literature [10]. Further, there have been no studies that have examined the time-course of strength maintenance following cross-education, thus we sought to investigate this. If cross-education is effective, then we would hypothesize that pain would reduce and function would improve, thus leading to a change in overall physical activity of participants. Consequently, we were also interested in examining the temporal effects of cross-education on strength maintenance.

## 2. Materials and Methods

### 2.1. Participants

Unilateral KOA participants (*n* = 26) and aged-matched healthy controls (*n* = 12) were recruited via the local hospital orthopedic clinic and local advertising. Prospective participants were required to have: (1) radiographic evidence of unilateral tibiofemoral knee osteoarthritis (Kellgren and Lawrence grade with a severity classification of 3–4); (2) independently living; (3) English speaking; (4) have a BMI of 20 to 35; and (5) able to provide informed consent.

Healthy age-matched control participants were required to: (1) be asymptomatic for knee or hip OA, as determined by radiograph evidence or lack of significant joint pain; (2) not be currently engaged in a strength training program; (3) independently living; (4) English speaking; (5) have a BMI of 20 to 35; and (6) be able to provide voluntary informed consent.

Participants were excluded for the following: (1) any participant unable to obtain medical clearance (uncontrolled hypertension, diabetes and angina); (2) evidence of bilateral KOA or hip OA; (3) a history of neurological disease or neurodegenerative conditions; (4) previous partial or complete knee or hip replacement to either leg; and (5) any form of cognitive impairment. The study was approved by the Deakin University Human Research Ethics Committee (DUHREC, ID: 2012-230).

### 2.2. Study Settings

KOA participants were recruited from an orthopedic ward of a local hospital, servicing both private and public health. Advertising was also utilized within the locale of Deakin University, Melbourne, Australia, to additionally recruit healthy control participants and KOA participants. Potential participants, who were identified as being unable to participate in the study due to lack of transportation, were offered transport in university fleet vehicles to and from the university for all testing and training sessions. All exercise (intervention) sessions were conducted in a university rehabilitation clinic by the same allied health professional. All assessment sessions were conducted in a physiology laboratory located in the same building.

### 2.3. Experimental Design

This was a non-blinded randomized control trial, with a 4-week intervention, with pre, post and a follow up assessment (3 months post intervention). Outcome measures of isometric knee extensor strength, rectus femoris muscle thickness and measures of neuromuscular function were assessed at all-time points.

A medical professional (orthopedic surgeon) based in the ward of the recruiting hospital assessed the eligibility of the potential participant via bilateral knee radiographs and determined medical clearance for potential participants, discussed the trial and ultimately recruited participants. Potential participants were then contacted by a researcher at the university (WB) to confirm interest in enrolment in the trial, to confirm informed consent, and to determine a suitable date for the initial assessment. As part of the initial assessment, Visual Analog Scale (VAS) and The Knee Injury and Osteoarthritis Outcome Score (KOOS) for pain were utilized to ensure no pain in the contralateral knee during functional tasks. Post assessment, an independent research fellow determined allocation to either the intervention or control groups. Healthy controls contacted the University directly to discuss the requirements of the trial and to determine suitability and provided informed consent. KOA participants were allocated into a unilateral exercise group or a non-exercise control group, via simple randomization with a 1:1 allocation. Immediately following the pre-assessment, a research fellow that was independent to the study and blinded to all attributes of the participant, determined the allocation of each participant. Allocation was implemented via a random number generator. Due to practical limitations, blinding of participants was not possible. The healthy age-matched controls were not randomized.

All participants in the exercise group participated in 12 supervised exercise sessions (3 per week for 4 weeks) of approximately 30 min. All exercise sessions were supervised by an experienced allied health professional. The post exercise testing occurred 3 days following the final training session. The 3-month follow up assessment occurred as close to 12 weeks as practically possible. Participants were asked to maintain their current physical activities throughout the duration of the study and not to commence any form of new physical activity, sport or exercise. This was assessed via a simple diary format which was assessed prior to each supervised training session.

### 2.4. Maximal Strength Testing

Maximal voluntary isometric contraction (MVIC) of the knee extensors and knee flexors was measured pre, immediately post intervention and at 3 months follow-up, with the unaffected contralateral limb tested prior to the KOA limb at each assessment. The participants were seated with their knees flexed at 60 degrees (−30 degrees from full knee extension) and the hip joint at 85 degrees on an isokinetic dynamometer (Biodex System 4 Pro, Biodex Medical Systems, Shirley, NY, USA), which has an intraclass correlation coefficient (ICC) of 0.93 for knee extension and an ICC of 0.89 for knee flexion [27]. Knee flexion of 60 degrees was selected to ensure consistency between testing sessions and participants; 60 degrees of knee flexion results in the greatest force output following and bypassing any potential restriction in movement due to KOA. The researcher instructed the participant to kick (extension) or pull (flexion) “as hard as possible” for 3-s, with three trials being performed, with a 2 min rest between each trial to minimize the effect of fatigue. Knee pain was measured via a VAS scale immediately following each trial in order to measure the potential influence of pain of the affected limb on knee extensor/flexor strength. Verbal encouragement was provided by the researchers and visual feedback of the force exerted was provided on the Biodex monitor which was located at eye level approximately 1 m from the participant. The raw force measured by the dynamometer was recorded in newton meters (NM) and was also normalized to each participant’s weight in kilograms (NM/kg).

### 2.5. Recording of Surface Electromyography

Surface electromyography (sEMG) was recorded from the rectus femoris (RF) muscle in both legs using Ag-AgCl electrodes. Two electrodes were placed 20 mm apart on the midpoint of the belly of RF, with the ground electrode placed on the lateral epicondyle of the tibia. Skin was prepared (shaven and cleaned with 70% isopropyl alcohol swabs) prior to the placement of the electrodes to ensure a clear signal was obtained. sEMG signals were amplified (1000×) with bandpass filtering between 20 Hz and 1 kHz and digitized at 10 kHz for 1 s, recorded and analyzed using PowerLab 4/35 (ADinstruments, Sydney, Australia). In a similar manner, to determine the extent of co-activation, sEMG was also recorded from the RF and biceps femoris (BF) in both legs. In relation to the BF, the muscle belly was identified via palpation during forceful knee flexion and correct placement was achieved by following the SENIEM guidelines and confirmed by examination of the sEMG activity during active internal and external rotation of the flexed knee.

### 2.6. Measurement of Muscle Thickness

We have previously reported excellent reliability for determining muscle thickness of the rectus femoris (RF) muscle (*r* = 0.99) using real-time ultrasound [28]. Therefore, using the same technique, a Nemio20 (Duluth, GA, USA) premium compact ultrasound was used to measure the thickness of the participant’s quadriceps muscle (RF) of each leg pre and post intervention. Measurements for muscle thickness were taken at the beginning of all testing sessions to ensure that exercise-induced changes in muscle blood flow did not affect the measurement. All measurements were performed by the same researcher (WB), with intra-experimenter coefficient of variation [CV] being between 2.6 and 3.8%.

The site of measurement was determined by marking the skin midway between the superior aspect of the patella and the anterior superior iliac spine, while the participant was in a supine position with the knee and hip in the anatomical position. The 6–8 Hz transducer probe was lubricated with transmission gel and placed lightly on the marked area of the skin. When a clear image was seen on the monitor, the pressure of the transducer to the skin was slowly reduced to ensure minimal compression of the muscle before the monitor was frozen. A cursor then marked the distance between the femur and the most superficial point of the muscle fascia, giving a distance which represented the thickness (mm) of the muscle under the marked point on the skin. Six readings were taken on each leg and averaged to determine the final value.

### 2.7. Interventions

All training sessions were supervised and monitored by an accredited exercise physiologist, with verbal encouragement given during in set. Participants completed a warm-up that consisted of two sets of unilateral leg press (Synergy Fitness, Omni Leg Press, Sydney, Australia) of the contralateral limb at progressively heavier loads (40% and 65% of one-repetition maximum [1RM]). The training consisted of four sets of 6–8 repetitions of unilateral leg press of the contralateral limb at >80% 1RM. This load was initially based on an 8RM unilateral strength measurement of the trained non-affected KOA limb determined during the initial training session. All participants were familiarized with the technique required prior to the first training session, with the focus of the initial training session on correct exercise technique to ensure no adverse effects, such as delayed onset muscle soreness or joint pain and swelling. A 3 min recovery period occurred between each set. Participants were required to perform each repetition with a repetition timing of 3 s of concentric and 4 s of eccentric; timing was measured by a metronome. The leg press was adjusted for each participant to ensure that the knee reached a minimum of 90 degrees as measured by a goniometer (3600 Baseline™ evaluation instruments, OPC Health, Melbourne, Australia). The principle of progressive overload was employed throughout the training period to maximize the training response [29]. Specifically, when participants could complete four sets of 8 repetitions, at the beginning of the next training session, the training weight (kg) was increased. All participants completed the 12 training sessions over the 4-week period.

### 2.8. Data Analysis

Participants were required to push or pull against (knee) the dynamometer and produce a gradual rise in force to its maximum over a 3 s interval. Once the maximum force was obtained it was held for a subsequent 3 s. Verbal encouragement and visual feedback of the force exerted was provided via a computer screen which was located at eye level approximately 1m away from the participant. MVIC was determined as the highest force (NM) recorded from three individual contractions. Knee extensor muscle girth (mm) was measured as the mean value from 6 individual recordings.

The contralateral transfer of strength was quantified using a procedure published by Carroll et al. [30] The magnitude of the cross-education effect was calculated as the mean change of knee extensor strength of the KOA intervention group to the untrained limbs of the KOA control group.
$$(\frac{EPost-EPre}{EPre}-\frac{CPost-CPre}{CPre})\times 100.$$

EPost referred to the mean repetition max (RM) of the experimental groups untrained knee extensors post intervention. EPre referred to the mean RM of the experimental groups knee extensors pre-intervention. CPost referred to the mean RM of the control groups untrained knee extensors post intervention. CPre referred to the mean RM of the control groups knee extensors pre-intervention.

The extent of hamstring co-activation was quantified using a procedure published by Hortobagyi [31]. The magnitude of co-activation was calculated as the percentage of maximal BF root mean square (RMS) EMG recorded during knee extension MVIC, compared to the maximal BF RMS EMG recorded during knee flexion MVIC.
Co-activation = (BF/BFmax)/(BF/RF) × 100 

Peak RMS EMG of BF was recorded during a knee flexion MVIC; the peak RMS EMG for BF was also recorded during knee extension MVIC. The BF/BF_max_ ratio, expressed as a percentage of total activation was then used to correctly interpret the extent of BF/RF ratio.

### 2.9. Statistical Analysis

We based our power calculations (G*Power, V3.1) on a meta-analysis that examined the effect of cross-education on knee extensor strength in healthy populations [32]. Based on a knee extensor cross-education effect of 10.4% (standard deviation [SD] ± 7.6), an a priori power analyses with a two-tailed *p*-value of 0.05 and a power of 0.95 (effect size [ES] 1.31) was conducted and we estimated that 10 participants was the minimum requirements for each group. While previous KOA studies have reported low dropout rates, to ensure adequate power, we adjusted recruitment to 16 participants per group [33].

Prior to statistical analysis, normality was screened with Shapiro–Wilk and Kolmogrov-Smirnov tests. If the data was not normally distributed, frequency histograms and detrended Q-Q plots were examined to determine if non-parametric tests were needed. If the data appeared normally distributed, a repeated measure analysis of variance (ANOVA) was used to determine the effect of the intervention on the dependent variables of knee extensor strength, knee flexor strength, and muscle thickness between the control and OA groups. Bonferroni post hoc test was performed on all possible comparisons to analyze any significant main effects and interactions. All dependent variables were tested for non-sphericity using Mauchly’s test. Any dependent variable not meeting the assumption of sphericity was adjusted by using the Greenhouse–Geisser correction. Correlation analysis was also used to examine any relationship between changes in muscle strength of the trained and untrained limbs [(post-strength/pre-strength ×100) − 100]. Significance level was set at *p* < 0.05 for all comparisons and all group data were provided as mean (M) ± SD in figures and as 95% confidence intervals in text.

### 2.10. Participant Flow

The final numbers analyzed were 16 KOA intervention, 10 KOA controls, and 12 healthy controls. Of the 74 KOA participants who expressed interest in the study, 28 were excluded for not meeting the requirements, bilateral KOA or other relevant medical issue, 5 declined to participate without reason and 11 due to distance or time. All participants in the KOA intervention group finished the study, and a further 4 participants allocated to the KOA control group were lost prior to post intervention assessment due to medical issues or illness.

Of the 20 healthy controls who expressed interest in the study, 12 healthy controls completed the study, with 2 not meeting the criteria with previous lower limb surgery, 1 declined participation, and the remaining 5 had subsequent medical issues prior to the initial assessment (i.e., stroke, heart attack, deceased or limb fracture). See the participant flow graph for further details (Figure 1).

## 3. Results

### 3.1. Baseline Characteristics

Twenty-six participants aged 55–76 years with radiographically diagnosed unilateral knee osteoarthritis (KL grade > 3) and 12 healthy age-matched controls were studied. There were no differences between groups for any characteristics including: age (*p* = 0.736), height (*p* = 0.834), weight (*p* = 0.703) and BMI (*p* = 0.869) (Table 1).

### 3.2. Trained Limb Knee Extensor Strength

At baseline, there were no differences in the strength of the knee extensors for the trained limb between the KOA intervention group and the KOA control group (*p* > 0.999). However, at baseline, there was a significant difference in strength of the trained knee extensors between the KOA intervention group and the healthy age-matched control group (*p* = 0.033). Further, there was also a significant difference at baseline in the strength of the trained knee extensor between the KOA control group and the healthy age-matched controls (*p* = 0.042, Figure 2).

Following the 4-week strength training intervention, there was a main effect for time (*F* _(1, 35)_ = 18; *p* < 0.001) and a group × time interaction (*F* _(2, 35)_ = 17; *p* < 0.001). Bonferroni post hoc analysis revealed that, for the trained limb in the KOA intervention group, maximum strength of the knee extensors increased by 24% (*p* < 0.001; *M* = 24, 95% *CI* [17, 32]), and this increase was significantly different to the KOA control group (*p* = 0.028, Figure 2). Importantly, the magnitude of change in knee extensor strength between the KOA intervention group and the healthy age-matched control group was not different (*p* = 0.16; Figure 2).

### 3.3. Untrained Limb Knee Extensor Strength

At baseline, there were no differences in the strength of the knee extensors for the untrained limb between the KOA intervention group and the KOA control group (*p* > 0.999). However, at baseline, there was a significant difference in strength of the knee extensors between the untrained KOA group and the healthy age-matched control group (*p* < 0.001). Further, there was also a significant difference at baseline in the strength of the knee extensor between the KOA control group and the healthy age-matched controls (*p* < 0.001).

Following the 4-week strength training intervention, there was a main effect for time (*F* _(1, 36)_ = 11; *p* < 0.001) and a group × time interaction (*F* _(2, 35)_ = 17; *p* < 0.001). Bonferroni post hoc analysis revealed that, for the untrained limb in the KOA intervention group, maximum strength of the knee extensors increased by 20% (*p* < 0.001; *M* = 15.3, 95% *CI* [8.8, 21.7]); further, this increase was significantly different to the KOA control group (*p* < 0.001). In addition, there was a significant difference following the intervention between the untrained limb in the KOA intervention group compared to the healthy age-matched control group (*p* = 0.001); Figure 3).

### 3.4. Contralateral Transfer of Strength

Unilateral strength training of the unaffected contralateral limb in unilateral KOA resulted in a cross transfer of strength of 19.8% to the untrained affected KOA limb, which equated to 78.2% of the strength gained in the trained limb. There was no relationship between the strength gained in the trained knee extensors and the contralateral transfer of strength to the untrained knee extensors (*r* = 0.42; *p* = 0.178; Figure 4).

### 3.5. Retention of Knee Extensor Strength for the Trained and Untrained Limb

Following the 3-month wash out period post intervention, there was no main effect for time observed (*F* _(1, 53)_ = 0.033; *p* = 0.857) or any group × time interactions (*F* _(2, 53)_ = 0.53; *p* = 0.593). No changes in knee extensor strength were observed in the 3 months following the intervention for the unaffected trained limb (*p* > 0.999; *M* = −4.8, 95% *CI* [−22, 13]) and the healthy age-matched control group (*p* > 0.999). Retention of knee extensor strength in the trained limb had occurred over the 3 months following the intervention.

For the untrained limb, 3 months post intervention, there was no main effect for time (*F*_(1, 53)_ = 0.033; *p* = 0.857) or any group × time interactions (*F*_(1, 53)_ = 0.53; *p* = 0.593). No changes in knee extensor strength were observed in the 3 months following the intervention for the affected untrained limb (*p* > 0.999; *M* = 2.5, 95% *CI* [−15, 20]) and the healthy age-matched control group (*p* > 0.999). A significant difference between the untrained limb of the KOA intervention group and KOA control group remained (*p* = 0.013, Figure 5). Retention of knee extensor strength in the affected untrained limb had occurred over the 3 months following the intervention.

### 3.6. Muscle Thickness for the Trained and Untrained Limb

At baseline, there were no differences in muscle thickness of the RF for the unaffected trained limb between the KOA intervention group and the KOA control group (*p* > 0.999). Furthermore, there were no differences in muscle thickness of the RF for the unaffected trained limb between the KOA intervention group and the healthy control group (*p* > 0.999). Again, there were no differences in muscle thickness of the RF for the trained limb between the KOA control group and the healthy control group (*p* > 0.999). Following the 4-week strength training intervention, there was a main effect for time (*F* _(1, 32)_ = 8.5; *p* = 0.006), however, there was no group × time interaction (*F*
_(2, 32)_ = 0.08; *p* = 0.923). Bonferroni post hoc analysis revealed that this change over time was not significant (4.1%, *p* = 0.317; *M* = −0.16, 95% *CI* [0.4, 0.08], see Appendix A).

Fort the untrained limb, there were no differences in muscle thickness of the RF for the affected untrained limb between the KOA intervention group and the KOA control group at baseline (*p* > 0.999). Furthermore, there were no differences in muscle thickness of the RF for the affected untrained limb between the KOA intervention group and the healthy control group (*p* > 0.999). Again, there were no differences in muscle thickness of the RF for the untrained limb between the KOA control group and the healthy control group (*p* > 0.999). Following the 4-week strength training intervention, there was a main effect for time (*F* _(1, 41)_ = 7.3; *p* = 0.036), however, there was no group × time interaction (*F* _(2, 41)_ = 0.041; *p* = 0.96). Bonferroni post hoc analysis revealed that, for the affected untrained limb in the KOA intervention, quadriceps muscle thickness of the untrained limb increased by 3.8% (*p* = 0.486; *M* = −0.13, 95% *CI* [−0.45, 0.19], see Appendix A).

### 3.7. Changes in Co-Activation for the Trained and Untrained Limb

At baseline, there were no differences in hamstring co-activation for the unaffected trained limb between the KOA intervention group and the KOA control group (*p* > 0.999). At baseline, there were no differences in hamstring co-activation between the KOA intervention group and the healthy age-matched control group (*p* = 0.994). At baseline, there were no differences in hamstring co-activation between the KOA control group and the healthy age-matched control group (*p* = 0.994). Further, there were no differences in hamstring co-activation between the KOA and unaffected trained limb in the KOA intervention group (*p* = 0.627).

Following the 4-week strength training intervention, there was no main effect for time (*F* _(1, 36)_ = 0.58; *p* = 0.451) but a group by time interaction occurred (*F* _(2, 36)_ = 3.8; *p* = 0.032). Bonferroni post hoc analysis revealed that, for the trained limb in the KOA intervention group, hamstring co-activation decreased by 12% (*p* = 0.655; *M* = 3.1, 95% *CI* [−0.77, 6.9]). Further, this decrease was significantly different to the KOA control group (*p* = 0.019). There was no significant difference following the intervention between the unaffected trained limb in the KOA intervention group compared to the healthy age-matched control group (*p* > 0.999). However, a group by time interaction occurred, with the KOA intervention group reducing hamstring co-contraction compared to KOA control group (*p* = 0.032, Figure 6).

For the untrained limb, at baseline, there were no differences in hamstring co-activation for the affected untrained limb between the KOA intervention group and the KOA control group (*p* > 0.999). However, there was a significant difference in hamstring co-activation between the KOA intervention group and the healthy age-matched control group (*p* = 0.008). Further, at baseline, there was a significant difference in hamstring co-activation between the KOA control group and the healthy age-matched control group (*p* > 0.006). Following the 4-week strength training intervention, there was no main effect for time (*F* _(1, 32)_ = 0.053; *p* = 0.819) but a group by time interaction occurred (*F* _(2, 32)_ = 5.3; *p* = 0.011). Bonferroni post hoc analysis revealed that, for the affected untrained limb in the KOA intervention group, hamstring co-activation decreased by 17.6% (*p* = 0.019; *M* = 6, 95% *CI* [0.8, 11]) and this decrease was significantly different to the KOA control group (*p* < 0.001, Figure 7).

## 4. Discussion

The purpose of this study was to determine the clinical efficacy of unilateral knee extensor training on imparting the cross-education phenomenon to the affected untrained limb in unilateral KOA. There were several important findings, which further validate the use of cross-education in unilateral KOA. The main findings were that there was a significant cross-education effect of 19.8% when the affected untrained limb of the KOA intervention group was compared to the affected untrained limb in the KOA control group. Retention of the knee extensor strength gains of both the trained and untrained KOA limbs was observed 3 months following the intervention, with the trained limb in the KOA intervention group showing no significant difference at both the post intervention and 3-month post intervention time point, confirming that the cross-transfer of strength was retained. In addition, changes in neuromuscular activation were observed in the affected untrained limb in the KOA intervention group compared to the KOA control, showing that cross-education modulates the neuromuscular system whereby antagonist co-contraction is reduced in the affected KOA limb.

### 4.1. Unilateral Strength Training Increase Strength of the Trained and Untrained Limb

Knee extensor strength of the trained limb significantly improved by 24%, with the untrained KOA limb improving by 20%. The magnitude of the cross-education effect was calculated by the method outlined by Carroll et al. [30]; 78% of the strength improvement of the trained limb transferred to the untrained KOA limb. The strength improvement of the untrained limb was greater than that seen in a recently published meta-analysis [12], which pooled data from the lower limb of 338 subjects, showing a mean improvement of 16.4%. While both distal and proximal muscles groups were included in the analysis, no differences were shown in the magnitude of the cross-education effect. While it appears the magnitude of cross-education was significantly greater in this study, there were several variables that could explain this. For example, in the meta-analysis by Manca et al. [12], the average age of the participants was 23.9 ± 3.3 years, whereas in this study the mean age of the KOA intervention group was 66.2 ± 5.6 years, a greater than 40-year difference. It is well-established that sedentary behavior becomes more common place as people age [34], and that the loss of muscle mass and strength is a normal part of the ageing process, with muscle strength declining more rapidly than muscle mass [35]. This could potentially lead to a large adaptive reserve, as a detrained individual is likely to make larger strength gains than a normal-to-well trained individual [36]. Supporting this, previous evidence showed that, in KOA, the loss of knee extensor strength is primarily neurological in nature and not due to muscle atrophy alone [5,37].

Previous studies have investigated cross-education in KOA, bilateral KOA and unilateral KOA [19]. These studies reported significant improvements in trained limb knee extensor strength, with Malas and colleagues reporting a 39.7% and Onigbinde and colleagues [19] reporting a 21% increase. Further, both reported significant improvements in knee extensor strength in the untrained limb of 27.3% and 21%, respectively. These were significantly higher than results from the previous meta-analysis [12]. Both Malas et al. [18] and Onigbinde et al. [19] utilized a highly specific isometric intervention that mimicked the testing procedure. The greater improvements in isometric knee extensor strength may potentially be explained due to a greater degree of familiarisation with the testing protocol [30]. Further, no control group was included in either the Malas et al. [18] or Onigbinde et al. [19] studies and neither study quantified the cross-education effect by the method outlined by Carroll [30]. This study also investigated cross-education in bilateral KOA, with no mention of the severity of the KOA for either limb. Conversely, the current study result can be considered more robust as participants had radiographic evidence of the grade of KOA and the use of a KOA control group allows the correct determination of the cross-education effect. Additionally, the intervention is more general in nature allowing greater transference into a community or clinical environment.

No significant changes in knee extensor muscle thickness were observed in either the trained or untrained contralateral limb in the KOA intervention group, from pre to post intervention. The cross-education phenomenon is a neurological adaptation [30], and while current research has suggested there is a possibility for muscle architecture changes [18], the majority of studies do not support this occurrence [11,16]. A strength of this study was the measurement of knee extensor muscle thickness, as a change in cross-sectional area (CSA) may impact the MVIC strength. This result supports the current research that the transfer of strength is wholly neurological in nature [10].

No changes in muscle thickness were observed in the KOA controls and the healthy controls from the pre to post time points, interestingly, there was no significant difference in muscle thickness between the limbs within each group and also between the three study groups as a whole. This suggest that in this population, the significant deficit in strength demonstrated in both the KOA intervention and KOA control groups when compared to the healthy control group, was not due to differences in muscle mass, but neurological factors. This is in support of previous research that demonstrated the loss of knee extensor strength in KOA is primarily due to the inability activate the muscle, not atrophy of the knee extensors [5,37].

### 4.2. Unilateral Strength Training Retains Strength of the Trained and Untrained Limb in KOA

Investigating the retention of strength improvements following a cross-education intervention is not a novel idea [38]. However, this is the first study that has investigated the retention of strength gains following a cross-education intervention in a clinical cohort. Previous detraining research in cross-education with a young healthy cohort has shown that, over 6 weeks, a significant decrease in strength of the trained limb occurred; but, interestingly, the untrained limb-maintained strength. This disparity possibly occurred as the trained limb had made significantly greater strength gains than the untrained limb in the young healthy cohort. Whereas, in the current study, the magnitude of the cross-education phenomenon was high, both limbs increased in knee extensor strength to a comparable degree. Further, when compared to healthy age-matched controls, the KOA intervention group in this study were significantly weaker prior to the intervention. It is possible that knee extensor strength was maintained as the KOA intervention group returned to normal levels of strength as seen in the healthy age-matched controls; whereas, if they had achieved knee extensor strength significantly higher than the controls some loss might have been observed.

Previous detraining research in the same age subjects (64.4 ± 0.9) has shown that strength loss in the lower limbs occurred at the same rate as younger adults and as little as a 9% decrease in a 6-month period following their intervention [39]. While a shorter detraining period of 2 months also displayed a 9% loss in strength, the participants were older (60–80 yrs old), comparably to the current study, suggesting that older adults may lose strength faster [40]. In comparison to cohorts of similar age, the retention of knee strength is impressive; however, the explanation may be simple.

Individuals with KOA do less incidental physical activity and less vigorous physical activity [41]. Following the intervention, it is possible that incidental physical activity and a return to normal recreational physical activity occurred in the KOA intervention group. This activity may have been the primary stimulus for retaining improvements made to knee extensor strength. However, a major limitation in this study is physical activity was not measured pre or post intervention, so these claims cannot be substantiated.

### 4.3. Unilateral Strength Training Reduces Hamstring Co-Activation in the Untrained Limb in KOA

Hamstring co-activation has previously been identified as an underlying pathology leading to decreased knee extensor strength in KOA [25,31,42,43]. Hamstring co-activation is thought to be a compensatory mechanism in KOA to add stability to the knee joint [44,45]. However, the trade-off for increased knee stability is knee extensor weakness. Prior to knee extension occurring, the quadriceps must overpower the torque created by the hamstring muscle group, reducing absolute knee extensor strength [46], and this leads to a decrease in walking speed in KOA cohorts [47].

Typically, hamstring co-activation studies in KOA are measured during a walking task in which the co-activation increases with increased gait speed, with the KOA cohort having significantly greater co-activation at all walking speeds when compared to healthy age-matched controls [25]. However, the current study measured co-activation during a MVIC, which paradoxically resulted in similar levels of co-activation compared to previous studies, where logically higher levels of force or sheering through the knee joint were expected to be observed [25,31]. Potentially, an MVIC in a seated and stable position results in no greater instability of the knee over a brisk walking task, hence the similarity between results.

Previous research has investigated a 4-week cross-education intervention in young healthy adults and hamstring co-activation during maximal isometric knee extension [46]. Interestingly, Tillin et al. [46] reported a different result from the current study, with no significant change in hamstring co-activation of the trained limb or the untrained limb, whereas the current study demonstrated a significant change in the untrained KOA limb, but no change in the trained limb. This difference may be explained as the underlying pathology in the KOA limb, resulting in greater initial levels of co-activation, especially when the significant difference to the healthy age-matched controls prior to the 4-week intervention is considered. Tillin et al. [46] postulated that the change in hamstring co-activation was not due to a decrease in hamstring activation, but an increase in knee extension force, therefore changing the percentage ratio. The current study has potentially shown the same result, with no changes pre to post intervention in hamstring EMG during knee extension MVICs.

Regardless of the mechanisms mediating the extent of hamstring co-activation during an MVIC, this is the first study, to the best of the authors’ knowledge that has shown changes in co-activation in an untrained limb following a cross-education intervention in participants with unilateral KOA.

### 4.4. Study Limitations

This study has several limitations that needed to be considered. Firstly, no assessment of socioeconomic status or previous occupational status were investigated, no blinding occurred, other than data analysis and group allocation, due to practical limitations. Secondly, only isometric strength measured during testing sessions and an 8-RM during the first and last training sessions were taken due to safety and ease of familiarisation in this elderly cohort. Thirdly, while physical activity was monitored during the intervention period to ensure no changes to normal physical activity, no measurement of baseline physical activity was made prior to the intervention period, or in the three months following the intervention, prior to the final assessment. This limitation meant that we could only theorize as to the reasons for the complete retention of strength gains in the intervention group. Due to practical limitations, there was no 3-month post intervention measurement of the KOA control group and no intention to treat analysis, which made it impossible to compare the KOA intervention to an equivalent control group at this time point. However, the novelty of measuring the 3-month detraining period in the KOA intervention and being able to compare them to a healthy control overcomes this limitation to a degree. Lastly, the relatively small sample size may limit the overall generalizability of the findings, particularly when taking into consideration the nine participants who withdrew from the study due to unrelated medical reasons.

## 5. Conclusions

This study supported our hypothesis that four weeks of strength training of the contralateral limb in unilateral KOA would result in a significant increase in strength to both the trained and untrained limb, and that the cross-education phenomenon would occur at a greater magnitude than in previous studies in young healthy subjects. The improvement in knee extensor strength of both the trained and untrained limbs was maintained for three months following the intervention. The cross-education phenomenon as an exercise therapy in improving bilateral knee extensor strength in this cohort was effective. Further, four weeks of strength training of the contralateral limb in unilateral KOA resulted in a reduction in hamstring co-activation.

## Figures and Tables

**Figure 1 jfmk-07-00077-f001:**
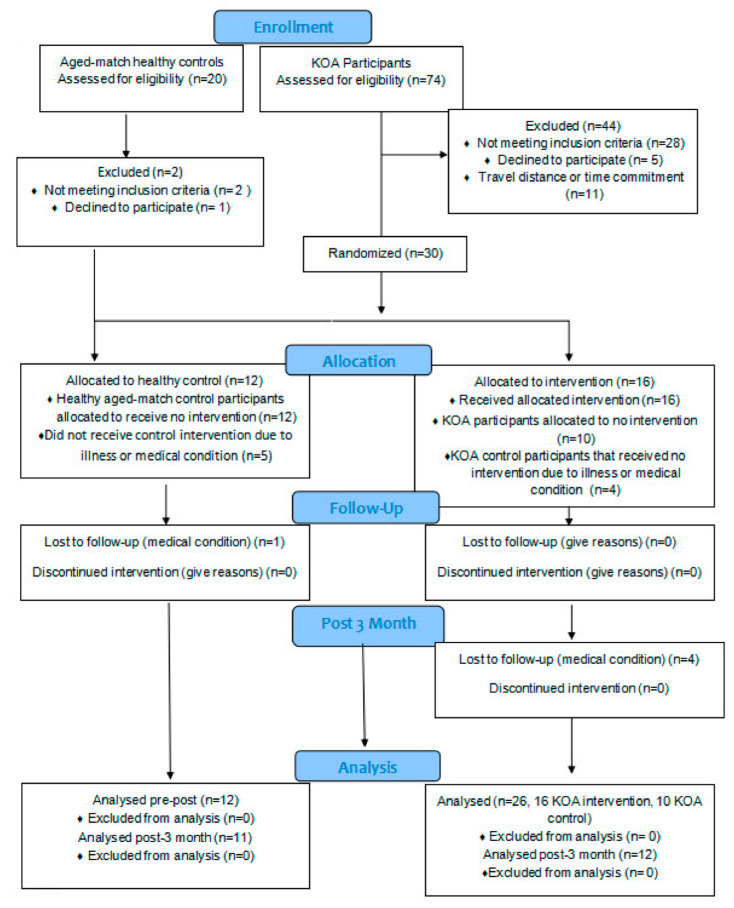
Consort flow chart of participant recruitment.

**Figure 2 jfmk-07-00077-f002:**
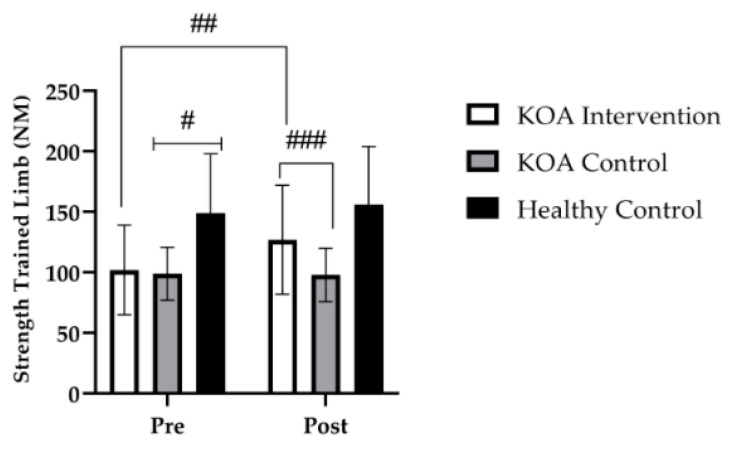
Group mean (±SD) data showing knee extensor strength of the trained limb. # denotes significant baseline differences of *p* < 0.001, between the healthy control group to the KOA intervention and KOA control. ## denotes significant time effect of *p* < 0.001, from baseline to post intervention for the KOA intervention. ### denotes a significant group by time interaction of *p* < 0.001 to the KOA controls and heathy control groups.

**Figure 3 jfmk-07-00077-f003:**
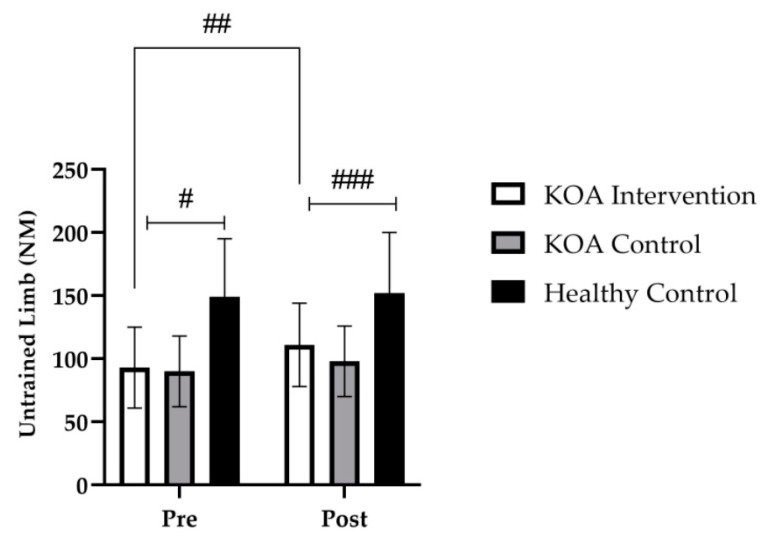
Group mean (±SD) data showing knee extensor strength of the untrained limb. # denotes significant baseline differences of *p* < 0.001, between the healthy control group to the KOA intervention and KOA control. ## denotes significant time effect of *p* < 0.001, from baseline to post intervention for the KOA intervention. ### denotes a significant group by time interaction of *p* < 0.001 to the KOA intervention and KOA controls to the heathy control group.

**Figure 4 jfmk-07-00077-f004:**
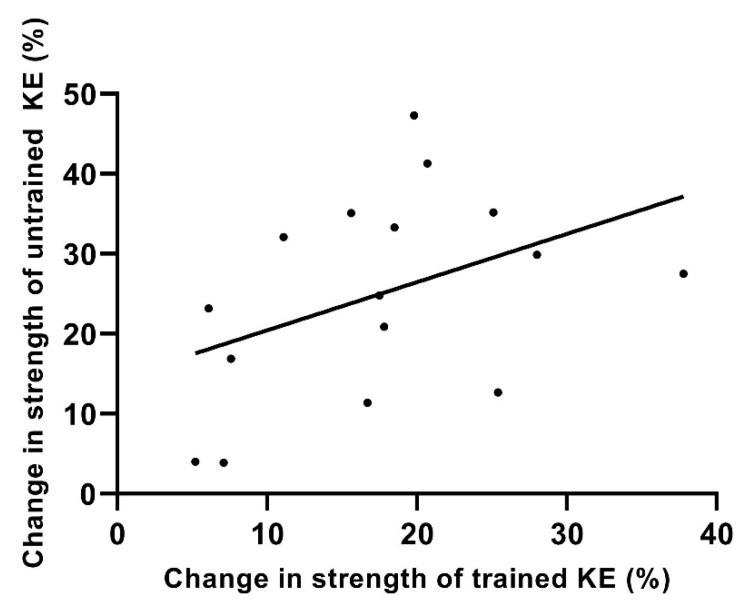
Strength changes for KE (knee extensor) 1-RM strength for the trained and untrained contralateral leg in KOA trained participants following 4 weeks of unilateral strength training. Data are expressed as a percentage of pretraining strength (*r* = 0.42; *p* = 0.178).

**Figure 5 jfmk-07-00077-f005:**
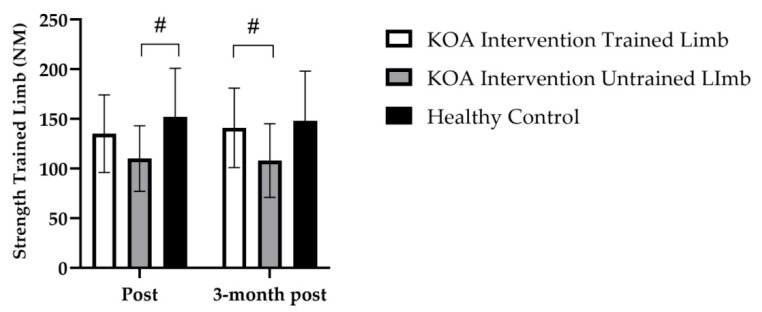
Group mean (±SD) data demonstrating the retention of the improvements in knee extension strength in the three months following the intervention. # denotes significant difference in strength between the KOA untrained limb and KOA control group at 3 months and a difference in strength post between KOA control and the healthy control group.

**Figure 6 jfmk-07-00077-f006:**
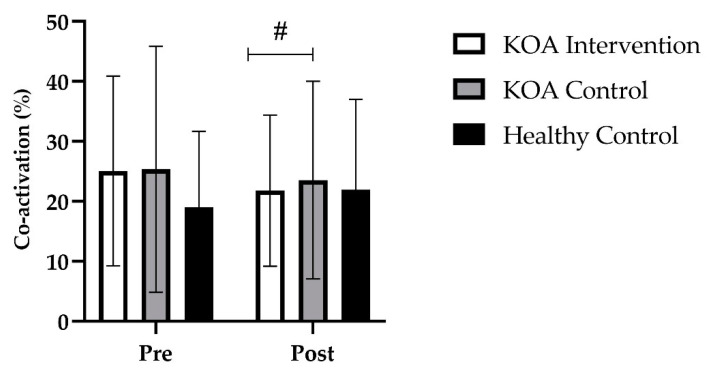
Group mean (±SD) data showing hamstring co-activation % of the trained limb. # denotes a significant group by time interaction of *p* < 0.05, between the KOA intervention and the KOA control.

**Figure 7 jfmk-07-00077-f007:**
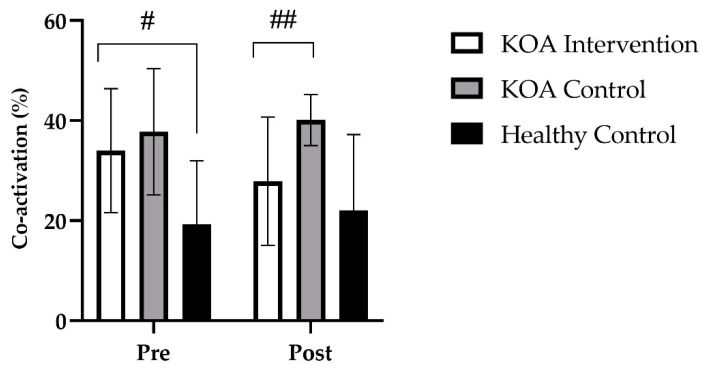
Group mean (±SD) data showing hamstring co-activation % of the untrained limb. # denotes a significant increase in co-activation at baseline between the KOA intervention and the KOA control group compared to the healthy control group. ## denotes a significant group by time interaction of *p* < 0.05, between the KOA intervention and KOA control group.

**Table 1 jfmk-07-00077-t001:** Values are mean ± SD. No significant differences between groups were observed.

	Age	Height (cm)	Weight (kg)	BMI	SEX
**KOA Intervention** **N = 16**	66.2 ± 5.6	169.9 ± 11.9	84.1 ± 12.7	29.1 ± 3.5	8 male, 8 females
**KOA Control** **N = 10**	63.7 ± 4	173.8 ± 9	88.2 ± 11.5	29.4 ± 3.3	5 male, 5 females
**Healthy Aged Match Control** **N = 12**	67 ± 6.9	171.1 ± 7.5	87.3 ± 17.9	29.7 ± 4.9	8 male, 4 females

## Data Availability

Not applicable.

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
