# Peer review of "Unilateral Strength Training Imparts a Cross-Education Effect in Unilateral Knee Osteoarthritis Patients"

_jfmk, 2022, doi:10.3390/jfmk7040077_

Round 1

Reviewer 1 Report

Thank you for inviting me to review the manuscript entitled "Unilateral Strength Training Imparts A Cross-education Effect in Unilateral Knee Osteoarthritis Patients." Overall, this is a well-written, scientifically sound, and basically interesting study for the readership of the Journal of Functional Morphology and Kinesiology. However, I have a few remarks, mostly concerning the results’ section, but these could well be addressed by the authors in a revision of the manuscript:

Introduction

Introduction: Well done, a very good read from my point of view. Just a few minor remarks:

Ll. 12-13: “Eighty-six million individuals over the age 20 were diagnosed with Knee 12 osteoarthritis (KOA) in 2020”. Please state which region your statement refers to. I guess it is worldwide…

Ll. 49 forward: The reader would certainly appreciate a brief explanation of the underlying physiological causes/processes of the cross-education phenomenon.

Ll. 51-52: The authors state that – among others – cross education has successfully been trialed in unilateral KOA [19] before, but its potential has yet to be explored. Please briefly elaborate on this. Which questions have not been answered yet by the study mentioned [ref#19], or other studies as well, which you try to answer with your trial.

Line 68: “completely innervate a muscle“ instead of “completely activate a muscle”?

Ll. 84-85: it says “and that the magnitude of improvement would be greater than 16.4% as seen in young healthy adults”. What is this hypothesis based on? It remains unclear to the reader, why the improvement in unilateral KOA is supposed to be greater than for healthy controls (although it is explained later on in the manuscript). Please briefly elaborate where appropriate (e.g., l. 50 onwards).

Ll. 85-87. Same as above. In the intro, no information is given regarding the temporal component of the effects. How long will the effects last. What are the corresponding results of the other studies (e.g. refs #13-21, particularly 18-19). The hypothesis on the persistence of the effects appears somewhat suddenly in this paragraph and without context, so that the three months follow-up seems arbitrarily chosen here. However, this could easily be cleared up by providing brief additional information in the intro.

Ll. 79-90: What is your hypothesis regarding muscle thickness (second outcome of your study)?

METHODS

L. 96: please define/explain KL grade (= Kellgren and Lawrence), as this abbreviation has not been mentioned before

Ll. 93 ff.: please indicate the number of participants included here

L. 120: please define VAS and KOOS, abbreviations have not been mentioned before

L. 149: Which limb was tested in the healthy control group/ how was it determined?

L. 152: Maybe rephrase “KOA limb” to “affected limb” here. Please re-check and use consistently throughout the manuscript.

Ll. 154-155: What was the sampling frequency of the isokinetic dynamometer?

L. 201: Please define 1RM here (has not been mentioned before).

L. 218-219: it says: “3-5 individual repetitions…”, but participants performed only three trials (compare l. 160). Please clarify!

L. 219: Were there objective indicators for determining a plateau? Or was the determination of the plateau based on subjective impressions (then please add this to the limitations of the study).

L. 242: It says: “a knee extensor cross-education effect of 10.4%”, while in line 85 (we hypothesized that) it is 16.4%. Please clarify.

RESULTS:

Overall, this section is quite extensive and difficult to read and follow. I think, this section would benefit from adding tables with all the detailed results. For readability purposes, these tables should replace figures 6-9 (which by the way could be added as supplementary materials), while the text in this section should only focus on significant results. This would facilitate working with the results of this trial in future systematic reviews and meta-analyses and also provide the reader with more detailed information (like means and mean changes, which are missing from the figures).

Ll. 279-280: It says “Twenty-eight participants aged 55-76 years with radiographically diagnosed unilateral knee osteoarthritis (KL grade >3) and 16 healthy age-matched controls were studied.” This contrasts with the number of participants presented in table 1. Additionally, in line 263-264 it says: “The final numbers analyzed were 16 KOA intervention, 10 KOA controls, and 12 healthy controls.” Please clarify!

L. 296: “…24% (p < 0.001; M = 24, 95% CI [17, 32]),” please define “M” = 24. Is it mean or median?

Ll. 298-299: “Importantly, the magnitude of change in knee extensor strength between the KOA intervention group and the healthy age-matched control group was not different”. As I understand it, the authors compare the delta for “pre vs. post”. There is a significant increase in knee extensor strength in the KOA intervention group. No increase was found among the KOA control group, therefore the delta differs significantly between KOA intervention and KOA control. However, it seems that almost no increase was found in the healthy control group. Therefore, I would assume that the delta between the KOA intervention and the healthy control group would differ significantly, too. Please re-check.

Ll. 330-336: Please compare the information presented in lines 330-332 with information in lines 334-336. Seems to me a repetition with different values (19.8% vs. 18%). Please clarify.

L. 337: Figure 4 represents individual date points of the KOA intervention group (trained & untrained limb). There seem to be only 15 instead of 16 data points. Please re-check. Also, a definition for the two lines in Figure 4 is missing.

Ll. 398-408: Without having access to the data and just looking at Figure 8, it seems that the information provided in this paragraph do not match the results presented in Figure 8. While hamstring co-activation is slightly reduced in KOA intervention and KOA control (intervention slightly more than KOA control), hamstring co-activation increased in the healthy control group. Therefore, I would assume a group*time interaction effect with significant differences between KOA intervention and healthy controls. This contrasts the results presented in Figure 8 on the right hand. Likewise, in line 404: shouldn’t it be: “in the KOA intervention group compared to the KOA control group” instead of “in the KOA intervention group compared to the healthy age-matched control”? Otherwise, the last two sentences of this paragraph would contradict each other. Please re-check!

Ll. 424-426: Again, it seems that the change (delta) between KOA intervention group and healthy control group differs (KOA intervention: delta = -17.6%; control group: delta = +xx%). However, comparing hamstring co-activation at post-assessment, KOA intervention and healthy controls do not seem to differ significantly. Please re-check/ clarify.

DISCUSSION

In general, in the discussion’s section, a paragraph dealing with the outcome muscle thickness is missing!

Ll. 432-445: Likewise, briefly refer to the results of your second outcome (muscle thickness). No significant changes have been observed – and wouldn’t have been expected. Nevertheless, as muscle thickness was assessed, it should be addressed in all the main sections of the manuscript (incl. the conclusion).

Ll. 518-519: Please add a reference for: “and this leads to a decrease in walking speed in KOA cohorts”

OVERALL:

Only minimal grammatical and spelling mistakes; but authors might want to re-check once again.

Again, this is an interesting manuscript that is well worth reading, and I look forward to the authors' revised version.

Author Response

Reviewer 1:

Comments and Suggestions for Authors

Thank you for inviting me to review the manuscript entitled "Unilateral Strength Training Imparts A Cross-education Effect in Unilateral Knee Osteoarthritis Patients." Overall, this is a well-written, scientifically sound, and basically interesting study for the readership of the Journal of Functional Morphology and Kinesiology. However, I have a few remarks, mostly concerning the results’ section, but these could well be addressed by the authors in a revision of the manuscript:

Response: We would like to thank the reviewer for their supportive and constructive comments regarding our manuscript. Where possible, we have tried to address all of the reviewer’s concerns.

INTRODUCTION

Introduction: Well done, a very good read from my point of view. Just a few minor remarks:

Ll. 12-13: “Eighty-six million individuals over the age 20 were diagnosed with Knee 12 osteoarthritis (KOA) in 2020”. Please state which region your statement refers to. I guess it is worldwide…

Response: We have corrected denoting world-wide on line 13 in red font.

Ll. 49 forward: The reader would certainly appreciate a brief explanation of the underlying physiological causes/processes of the cross-education phenomenon.

Response: We thank the reviewer for this comment and we have now added the following in red font from line 54-58. “The proposed neurological mechanisms that underpin the cross-education effect appear to be driven by changes in the excitability of the primary motor cortex ipsilateral to the trained limb [10]. Specifically increased ipsilateral corticospinal excitability, reduced corticospinal inhibition, reduced short-interval cortical inhibition and potentially reduced interhemispheric inhibition [10].”

Ll. 51-52: The authors state that – among others – cross education has successfully been trialed in unilateral KOA [19] before, but its potential has yet to be explored. Please briefly elaborate on this. Which questions have not been answered yet by the study mentioned [ref#19], or other studies as well, which you try to answer with your trial.

Response: We agree with the reviewer and we have modified this sentence and it now reads as follows “The application of cross-education in unilateral KOA appears to have merit; however previous studies [18, 19] did not examine the potential neuromuscular mechanisms that underpin the cross-education effect.

Line 68: “completely innervate a muscle“ instead of “completely activate a muscle”?

Response: Corrected as suggested. The sentence now reads “Decreased knee extensor strength throughout the progression of KOA is primarily due to the inability of the nervous system to completely innervate a muscle, which is termed central activation deficit (CAD) [23].

Ll. 84-85: it says “and that the magnitude of improvement would be greater than 16.4% as seen in young healthy adults”. What is this hypothesis based on? It remains unclear to the reader, why the improvement in unilateral KOA is supposed to be greater than for healthy controls (although it is explained later on in the manuscript). Please briefly elaborate where appropriate (e.g., l. 50 onwards).

Response: We agree with the reviewer and upon reflection, the reported hypothesis is incorrect, as the purpose of the study was to examine whether cross-education actually exists in a cohort with unliteral KOA. Thus, we have decided to revise the hypothesis. It now reads as follows “It was hypothesized that unilateral strength training of the unaffected contralateral limb in unilateral KOA would increase knee extensor strength bilaterally, imparting a cross-education effect to the untrained KOA limb”.

Ll. 85-87. Same as above. In the intro, no information is given regarding the temporal component of the effects. How long will the effects last. What are the corresponding results of the other studies (e.g. refs #13-21, particularly 18-19). The hypothesis on the persistence of the effects appears somewhat suddenly in this paragraph and without context, so that the three months follow-up seems arbitrarily chosen here. However, this could easily be cleared up by providing brief additional information in the intro.

Response: We thank the reviewer for this comment and we have tried to address this point with the following “. It was hypothesized that unilateral strength training of the unaffected contralateral limb in unilateral KOA would increase knee extensor strength bilaterally, imparting a cross-education effect to the untrained affected KOA limb. Further, the improvements in knee extensor strength would be retained in the three-month period following the intervention. It was also hypothesized that unilateral strength training of the unaffected contralateral limb in participants with unilateral KOA would decrease co-activation of the hamstring muscle group, underpinning any changes in strength in the trained and untrained limbs. Given that the reported neural mechanisms modulating the cross-education effect, the above hypotheses appear to be supported by the literature [e.g. 10]. Further, there have been no studies that have examined the time-course of strength maintenance following cross-education, thus we sought to investigate this. If cross-education is effective, then we would hypothesize that pain would reduce and function would improve, thus leading to a change in overall physical activity of participants. Thus, we were also interested in examining the temporal effects of strength maintenance, following cross-education.”

Ll. 79-90: What is your hypothesis regarding muscle thickness (second outcome of your study)?

Response: We thank the reviewer for this comment. The hypothesis would be that if cross-education was present in the affected limb, then muscle thickness would be mainlined. Please see Haggert et al 2021 (https://www.josam.org/josam/article/view/54)

METHODS

  1. 96: please define/explain KL grade (= Kellgren and Lawrence), as this abbreviation has not been mentioned before

Response: We have added as suggested (line 105) and the sentence now reads as “Unilateral KOA participants and aged-matched healthy controls were recruited via the local hospital orthopedic clinic and local advertising. Prospective participants were required to have: 1) radiographic evidence of unilateral tibiofemoral knee osteoarthritis (Kellgren and Lawrence grade with a severity classification of 3-4); 2) independently living; 3) English speaking; 4) have a BMI of 20 to 35; and 5) able to provide informed consent.

Ll. 93 ff.: please indicate the number of participants included here

Response: Added as suggested, line 103. It now reads as ‘Unilateral KOA participants (n = 26) and aged-matched healthy controls (n = 12) were recruited via the local hospital orthopedic clinic and local advertising. Prospective participants were required to have: 1) radiographic evidence of unilateral tibiofemoral knee osteoarthritis (Kellgren and Lawrence grade with a severity classification of 3-4); 2) independently living; 3) English speaking; 4) have a BMI of 20 to 35; and 5) able to provide informed consent.

  1. 120: please define VAS and KOOS, abbreviations have not been mentioned before

Response: Amended as suggested, please see line 126 “As part of the initial assessment, Visual Analog Scale (VAS) and The Knee Injury and Osteoarthritis Outcome Score (KOOS) for pain were utilized to ensure no pain in the contralateral knee during functional tasks.

  1. 149: Which limb was tested in the healthy control group/ how was it determined?

Response: We thank the reviewer for this comment, however, both limbs for the control group and KOA groups were tested throughout the study period.

  1. 152: Maybe rephrase “KOA limb” to “affected limb” here. Please re-check and use consistently throughout the manuscript.

Response: Corrected as suggested, please see line 165 “Knee pain was measured via VAS scale immediately following each trial, to measure the potential in-fluence of pain of the affected KOA limb on knee extensor strength”.

Ll. 154-155: What was the sampling frequency of the isokinetic dynamometer?

Response: It was 100Hz.

  1. 201: Please define 1RM here (has not been mentioned before).

Response: Corrected as suggested.

  1. 218-219: it says: “3-5 individual repetitions…”, but participants performed only three trials (compare l. 160). Please clarify!

Response: We thank the reviewer for noting this oversight. It was 3 trials, and we have amended accordingly.

  1. 219: Were there objective indicators for determining a plateau? Or was the determination of the plateau based on subjective impressions (then please add this to the limitations of the study).

Response: We added to this by including the following “Participants were required to push or pull against (knee) the dynamometer and produce a gradual rise in force to its maximum over a 3 s interval. Once the maximum force was obtained it was held for a subsequent 3 s. Verbal encouragement and visual feedback of the force exerted was provided via a computer screen which was located at eye level approximately 1.5 m away from the participant. MVIC was determined as the highest force (NM) recorded from three individual contractions”.

  1. 242: It says: “a knee extensor cross-education effect of 10.4%”, while in line 85 (we hypothesized that) it is 16.4%. Please clarify.

Response: We thank the reviewer for this comment, however, we have since removed the 16.4%, thus we feel we should retain the 10.4% as this is what was determined our a prior sample size calculation.

RESULTS:

Overall, this section is quite extensive and difficult to read and follow. I think, this section would benefit from adding tables with all the detailed results. For readability purposes, these tables should replace figures 6-9 (which by the way could be added as supplementary materials), while the text in this section should only focus on significant results. This would facilitate working with the results of this trial in future systematic reviews and meta-analyses and also provide the reader with more detailed information (like means and mean changes, which are missing from the figures).

Response: We would like to thank the reviewer for their comments on the results and whilst we agree the results are difficult to follow, we do believe we have reported them adequately. For example, we have reported the percentage change and the 95% Confidence interval which provides the reader with additional important information that the Figures do not contain. In addition, the Figures report the mean raw data along with SD, therefore we have provided a suitable representation of the results, showing mean changes, mean raw data along with measures of variance. We disagree with the reviewer regarding only reporting significant results as this increases reporting bias. Overall, we feel the results should be retained with both significant and non-significant findings. In addition, the raw data presented in the Figures, valuably support our significant and non-significant findings showing transparency of our results. We agree that figures 6 and 7 could be supplementary files, as such we have amended.

Ll. 279-280: It says “Twenty-eight participants aged 55-76 years with radiographically diagnosed unilateral knee osteoarthritis (KL grade >3) and 16 healthy age-matched controls were studied.” This contrasts with the number of participants presented in table 1. Additionally, in line 263-264 it says: “The final numbers analyzed were 16 KOA intervention, 10 KOA controls, and 12 healthy controls.” Please clarify!

Response: We thank the reviewer for identifying this oversight. The numbers were 16 KOA intervention, 10 KOA controls and 12 healthy aged-matched controls. We have now corrected the error and it now reads as follows” Twenty-six participants aged 55-76 years with radiographically diagnosed unilateral knee osteoarthritis-tis (KL grade >3) and 16 healthy age-matched controls were studied. There were no differences between groups for any characteristics including: age (p = 0.736), height (p = 0.834), weight (p = 0.703) and BMI (p = 0.869) (Table 1).

  1. 296: “…24% (p < 0.001; M = 24, 95% CI [17, 32]),” please define “M” = 24. Is it mean or median?

Response: We thank the reviewer for this comment and as outlined under the statistical analysis section, all data in text are reported as the Mean (M) and the corresponding 95% Confidence interval. We have highlighted this in red font in the statistical analyses section for clarity.

Ll. 298-299: “Importantly, the magnitude of change in knee extensor strength between the KOA intervention group and the healthy age-matched control group was not different”. As I understand it, the authors compare the delta for “pre vs. post”. There is a significant increase in knee extensor strength in the KOA intervention group. No increase was found among the KOA control group; therefore, the delta differs significantly between KOA intervention and KOA control. However, it seems that almost no increase was found in the healthy control group. Therefore, I would assume that the delta between the KOA intervention and the healthy control group would differ significantly, too. Please re-check.

Response: We thank the reviewer for this comment and we have re-checked our data and the reported information is correct. The reviewer is correct in that the healthy aged-matched controls did not differ from baseline as they were a control group and did not complete any strength training. Rather, the important finding here is that following the 4-week training intervention, the KOA intervention group increased their strength and this increase was similar to that of the healthy control group, thus removing the baseline difference. This is important because the non-affected limb in unilateral KOA patients is actually different to that of healthy controls, a point we discuss in the introduction and is supported by our baseline difference. At a minimum, our data show that a short unilateral strength training intervention brings KOA patients back up to the level of their healthy peers.

Ll. 330-336: Please compare the information presented in lines 330-332 with information in lines 334-336. Seems to me a repetition with different values (19.8% vs. 18%). Please clarify.

Response: We would like to thank the reviewer for their attention to detail here and as suggested this was an error on our behalf. The transfer was in fact 19.8%. We have subsequently removed the last sentence.

  1. 337: Figure 4 represents individual date points of the KOA intervention group (trained & untrained limb). There seem to be only 15 instead of 16 data points. Please re-check. Also, a definition for the two lines in Figure 4 is missing.

Response: Again, we thank the reviewer for their attention to detail. The reviewer is correct and there was a missing data point which we have now added. The results now read as “Unilateral strength training of the unaffected contralateral limb in unilateral KOA resulted in a cross transfer of strength of 19.8% to the untrained affected KOA limb, which equated to 78.2% of the strength gained in the trained limb. There was no relationship between the strength gained in the trained knee extensors and the contralateral transfer of strength to the untrained knee extensors (r = 0.42; p = 0.178; Figure 4). We have also corrected figure 4 for simplicity.

Ll. 398-408: Without having access to the data and just looking at Figure 8, it seems that the information provided in this paragraph do not match the results presented in Figure 8. While hamstring co-activation is slightly reduced in KOA intervention and KOA control (intervention slightly more than KOA control), hamstring co-activation increased in the healthy control group. Therefore, I would assume a group*time interaction effect with significant differences between KOA intervention and healthy controls. This contrasts the results presented in Figure 8 on the right hand. Likewise, in line 404: shouldn’t it be: “in the KOA intervention group compared to the KOA control group” instead of “in the KOA intervention group compared to the healthy age-matched control”? Otherwise, the last two sentences of this paragraph would contradict each other. Please re-check!

Response: We thank the reviewer again for citing this error. The reviewer is correct and the group*time effect was in fact between the intervention group and the KOA control group. We have amended as noted by the reviewer.

Ll. 424-426: Again, it seems that the change (delta) between KOA intervention group and healthy control group differs (KOA intervention: delta = -17.6%; control group: delta = +xx%). However, comparing hamstring co-activation at post-assessment, KOA intervention and healthy controls do not seem to differ significantly. Please re-check/ clarify.

Response: We thank the reviewer for citing this error. The reviewer is correct and the group*time effect was in fact between the intervention group and the KOA control group. We have amended as noted by the reviewer.

DISCUSSION

In general, in the discussion’s section, a paragraph dealing with the outcome muscle thickness is missing!

Response: We have added the following in red font to address this point. No significant changes in knee extensor muscle thickness were observed in either the trained or un-trained contralateral limb in the KOA intervention group, from pre to post intervention. The cross-education phenomenon is a neurological adaptation [30], and while current research has suggested there is a possibility muscle architecture changes [18], the majority of studies do not support this occurrence [11, 16]. A strength of this study was the measurement of knee extensor muscle thickness, as a change in cross sectional area (CSA) may impact the MVIC strength.  This result supports the current research that the transfer of strength is wholly neurological in nature [10].

No changes in muscle thickness were observed in the KOA controls and the healthy controls from the pre to post time points, interestingly, there was no significant difference in muscle thickness between the limbs within each group and also between the three study groups as a whole. This suggest that in this population, the significant deficit in strength demonstrated in both the KOA intervention and KOA control groups when compared to the healthy control groups, was not due to differences in muscle mass, but neurological factors. This is in support of previous research that demonstrated the loss of knee extensor strength in KOA is primarily due to the inability activate the muscle, not atrophy on the knee extensors [5, 37].”

Ll. 432-445: Likewise, briefly refer to the results of your second outcome (muscle thickness). No significant changes have been observed – and wouldn’t have been expected. Nevertheless, as muscle thickness was assessed, it should be addressed in all the main sections of the manuscript (incl. the conclusion).

Response: We have addressed accordingly now in the discussion.

Ll. 518-519: Please add a reference for: “and this leads to a decrease in walking speed in KOA cohorts”

Response: We have added the following reference:

  1. McAlindon, T.A., Cooper, C., Kirwan, J.R., Dieppe, P.A., Determinants of disability in osteoarthritis of theknee. Ann Rheum Dis, 1993. 52:528-62.

OVERALL:

Only minimal grammatical and spelling mistakes; but authors might want to re-check once again.

Response: We thank the reviewer for this comment and we have since had the revised manuscript proof-read by an editorial service. We hope that errors in syntax have now been corrected.

Again, this is an interesting manuscript that is well worth reading, and I look forward to the authors' revised version.

Reviewer 2 Report

Number of enrolled participants in this study are really small KOA intervention 16; KOA controls 10; healthy controls 12. It is suggested that there should be in the title "preliminary study"

Also numbers in rows 276-280 (KOA participants 28, healthy 16) are NOT corresponding with Table 1 where one can find following numbers (KOA interventions 16; KOA control 10; Healthy controls 12). This inaccuracy should be corrected.

One should define word significance in conclusion, as we all know (conventional wisdom) that exercises brings improvement.  So, better results for 20%  after 4 weeks of exercises is expected result, isn't it?  Why this result is called significant?

Reference 28 is not referring to ultrasound as quoted by authors, (see row 178-180): "We have previously reported excellent reliability for determining muscle thickness 179 of the rectus femoris (RF) muscle (r = 0.99) using real-time ultrasound [28]." This is inappropriate citation as this reference has abstract as follows (founded in library of Acta Physiologica 2012) with no word "ultrasound" in this abstract (see bellow)

Abstract

Aim

Paired-pulse transcranial magnetic stimulation was used to investigate the influence of 4 weeks of heavy load squat strength training on corticospinal excitability and short-interval intracortical inhibition (rectus femoris muscle).

Methods

Participants (n = 12) were randomly allocated to a strength training or control group. The strength training group completed 4 weeks of heavy load squat strength training. Recruitment curves were constructed to determine values for the slope of the curve, V50 and peak height. Short-interval intracortical inhibition was assessed using a subthreshold (0.7 × active motor threshold) conditioning stimulus, followed 3 ms later by a supra-threshold (1.2 × active motor threshold) test stimulus. All motor evoked responses were taken during 10% of maximal voluntary isometric contraction (MVC) torque and normalized to the maximal M-wave.

Results

The strength training group attained 87% increases in 1RM squat strength (P < 0.01), significant increases in measures of corticospinal excitability (1.2 × Motor threshold: 116%, P = 0.016; peak height of recruitment curve = 105%, P < 0.001), and a 32% reduction in short-interval intracortical inhibition (P < 0.01) following the 4-week intervention compared with control. There were no changes in any dependent variable (P > 0.05) detected in the control group.

Conclusion

Repeated high force voluntary muscle activation in the form of short-term strength training reduces short-interval intracortical inhibition. This is consistent with studies involving skilled/complex tasks or novel movement patterns and acute studies investigating acute voluntary contractions.

It is suggested that Reference 28 should be omitted or appropriately used.

Author Response

Reviewer 2:

Comments and Suggestions for Authors

Number of enrolled participants in this study are really small KOA intervention 16; KOA controls 10; healthy controls 12. It is suggested that there should be in the title "preliminary study"

Response: We would like to thank the reviewer for this comment, however we hope that the reviewer can appreciate the difficulty in conducting intervention trials with patients. We would also like to draw attention to the reviewer, that our sample size (38 in total) exceeds the average sample size of most cross-education studies (see Manca et al., 2017 for details). In fact, according to the elegant meta-analysis by Manca et al. (2017) the average total sample size was 21 subjects (12 cases, 9 controls). Thus, we feel we have recruited well beyond the average.

[1] 1. Manca, A., et al., Cross-education of muscular strength following unilateral resistance training: a meta-analysis. European journal of applied physiology, 2017. 117(11): p. 2335-2354.

Also numbers in rows 276-280 (KOA participants 28, healthy 16) are NOT corresponding with Table 1 where one can find following numbers (KOA interventions 16; KOA control 10; Healthy controls 12). This inaccuracy should be corrected.

Response: We thank the reviewer for their attention detail. As noted by reviewer 1, we have corrected this to the following ‘Twenty-six participants aged 55-76 years with radiographically diagnosed unilateral knee osteoarthritis-tis (KL grade >3) and 12 healthy age-matched controls were studied. There were no differences between groups for any characteristics including: age (p = 0.736), height (p = 0.834), weight (p = 0.703) and BMI (p = 0.869) (Table 1).’

One should define word significance in conclusion, as we all know (conventional wisdom) that exercises brings improvement.  So, better results for 20%  after 4 weeks of exercises is expected result, isn't it?  Why this result is called significant?

Response: We thank the reviewer for this comment, however, we feel that our use of significant is correct and doesn’t requiring defining. We base this upon our “a prior power” calculation which was set to show a 10.4% increase in strength of the untrained limb, and we have shown a 20% increase in strength in a population with arthrogenic muscle inhibition, whereby it is totally conceivable that the cross-education phenomena could have been blocked to due cortical mechanisms associated with AMI. Thus, we feel that the transfer is in fact significant for the population that we have studied.

Reference 28 is not referring to ultrasound as quoted by authors, (see row 178-180): "We have previously reported excellent reliability for determining muscle thickness 179 of the rectus femoris (RF) muscle (r = 0.99) using real-time ultrasound [28]." This is inappropriate citation as this reference has abstract as follows (founded in library of Acta Physiologica 2012) with no word "ultrasound" in this abstract (see bellow)

Abstract

Aim

Paired-pulse transcranial magnetic stimulation was used to investigate the influence of 4 weeks of heavy load squat strength training on corticospinal excitability and short-interval intracortical inhibition (rectus femoris muscle).

Methods

Participants (n = 12) were randomly allocated to a strength training or control group. The strength training group completed 4 weeks of heavy load squat strength training. Recruitment curves were constructed to determine values for the slope of the curve, V50 and peak height. Short-interval intracortical inhibition was assessed using a subthreshold (0.7 × active motor threshold) conditioning stimulus, followed 3 ms later by a supra-threshold (1.2 × active motor threshold) test stimulus. All motor evoked responses were taken during 10% of maximal voluntary isometric contraction (MVC) torque and normalized to the maximal M-wave.

Results

The strength training group attained 87% increases in 1RM squat strength (P < 0.01), significant increases in measures of corticospinal excitability (1.2 × Motor threshold: 116%, P = 0.016; peak height of recruitment curve = 105%, P < 0.001), and a 32% reduction in short-interval intracortical inhibition (P < 0.01) following the 4-week intervention compared with control. There were no changes in any dependent variable (P > 0.05) detected in the control group.

Conclusion

Repeated high force voluntary muscle activation in the form of short-term strength training reduces short-interval intracortical inhibition. This is consistent with studies involving skilled/complex tasks or novel movement patterns and acute studies investigating acute voluntary contractions.

It is suggested that Reference 28 should be omitted or appropriately used.

Response: We thank the reviewer fort his comment, However, we are unclear as to why it is not appropriate? We are simply stating that our research group has previously demonstrated excellent reliability for using ultrasonography to determine muscle thickness for the exact same muscle that we have used in the present study. Whilst the cited reference was a strength training TMS study, that particular paper clearly states the following that supports our statement within the current study:

Weier et al (2012) Acta Physiologica

“Specifically, eight participants were assessed on two consecutive days prior to the commencement of the intervention. An average of six recordings was obtained from each participant and subsequently used for reliability analysis. The testing procedure was found to be reliable, with no significant difference detected between the two testing sessions, and a coefficient of variation of less than 1% for the left (P= 0.81, r = 0.99) and right (P= 0.73, r = 0.99).

Thus, we are unclear as to how we have used this reference incorrectly when it is referring to the rectus femoris muscle thickness.

  1. Manca, A., et al., Cross-education of muscular strength following unilateral resistance training: a meta-analysis.European journal of applied physiology, 2017. 117(11): p. 2335-2354.

Reviewer 3 Report

Dear Authors,

it was a pleasure for me to review such a carefully prepared manuscript. I have only few minor comments:

- please provide the date of Ethics Committee approval;

_ I would suggest to move section 2.3. before section 2.2.;

- please explain the placement of sEMG electrodes on BF muscles in detail;

- fig. 1 - it seems, that there is a mistake in post 3 month box in the experimental group - it should be moved to the control group;

- it is not clear when you lost 4 participants from the control group - before the middle or the last follow-up? Please make it clear on the flow graph;

- additionally - the control group did not receive intervention, as I understand from the text; therefore, you should not write "received allocated intervention" on the graph; the same for healthy controls;

- you can consider including appendix with tables including detailed results (values) - this can help to increase citations; 

- line 468 - "Onigbinde and colleagues" should be "Onigbinde et al.";

Author Response

Reviewer 3:

Comments and Suggestions for Authors

Dear Authors,

it was a pleasure for me to review such a carefully prepared manuscript. I have only few minor comments:

Response: We would like to thank the reviewer for their supportive comments regarding our manuscript. We have addressed all comments raised by this reviewer.

- please provide the date of Ethics Committee approval;

Response: We have highlighted this in red font on line 117.

 I would suggest to move section 2.3. before section 2.2.;

Response: Moved as suggested

- please explain the placement of sEMG electrodes on BF muscles in detail;

Response: Added as suggested

- fig. 1 - it seems, that there is a mistake in post 3 month box in the experimental group - it should be moved to the control group;

Response: We would like to thank the reviewer for their attention detail. We believe the text and the associated figure was figure 5. We have corrected as noted.

- it is not clear when you lost 4 participants from the control group - before the middle or the last follow-up? Please make it clear on the flow graph;

Response: We thank the reviewer for this comment and we note that the consort diagram shows that 2 participants didn’t meet the inclusion criteria, 1 one participant declined and 1 participant was lost at follow up at 3 months. We feel that the consort chart adequately displays this.  However, in agreement with the reviewer we have simplified Figure 1 for clarity.

- additionally - the control group did not receive intervention, as I understand from the text; therefore, you should not write "received allocated intervention" on the graph; the same for healthy controls;

Response: We thank the reviewer for this comment and we have revised the consort chart.

- you can consider including appendix with tables including detailed results (values) - this can help to increase citations;

Response: We thank the reviewer this comment.

- line 468 - "Onigbinde and colleagues" should be "Onigbinde et al.";

Response: Corrected as suggested.

Round 2

Reviewer 1 Report

Obviously, the authors have substantially revised their manuscript according to the initial comments. Where no revisions have been made, I can follow the argumentation of the authors. Therefore, my recommendation is: Accept in present form.